# Regenerative Potential of the Product “CardioCell” Derived from the Wharton’s Jelly Mesenchymal Stem Cells for Treating Hindlimb Ischemia

**DOI:** 10.3390/ijms20184632

**Published:** 2019-09-18

**Authors:** Aleksandra Musiał-Wysocka, Marta Kot, Maciej Sułkowski, Marcin Majka

**Affiliations:** Department of Transplantation, Faculty of Medicine, Medical College, Jagiellonian University, Wielicka 265, 30-663 Kraków, Poland; aleksandra.musial@uj.edu.pl (A.M.-W.); marta.kot@uj.edu.pl (M.K.); maciek.sulkowski@uj.edu.pl (M.S.)

**Keywords:** mesenchymal stem cells, hindlimb ischemia, regeneration, WJ-MSCs

## Abstract

In recent years, mesenchymal stem cells (MSCs) have emerged as a promising therapeutic modality in regenerative medicine. They hold great promise for treating civilization-wide diseases, including cardiovascular diseases, such as acute myocardial infarction and critical limb ischemia. MSCs isolated from Wharton’s jelly (WJ-MSCs) may be utilized in both cell-based therapy and vascular graft engineering to restore vascular function, thereby providing therapeutic benefits for patients. The efficacy of WJ-MSCs lies in their multipotent differentiation ability toward vascular smooth muscle cells, endothelial cells and other cell types, as well as their capacity to secrete various trophic factors, which are potent in promoting angiogenesis, inhibiting apoptosis and modulating immunoreaction. Ischemic limb disease is caused by insufficient nutrient and oxygen supplies resulting from damaged peripheral arteries. The lack of nutrients and oxygen causes severe tissue damage in the limb, thereby resulting in severe morbidities and mortality. The therapeutic effects of the conventional treatments are still not sufficient. Cell transplantations in small animal model (mice) are vital for deciphering the mechanisms of MSCs’ action in muscle regeneration. The stimulation of angiogenesis is a promising strategy for the treatment of ischemic limbs, restoring blood supply for the ischemic region. In the present study, we focus on the therapeutic properties of the human WJ-MSCs derived product, Cardio. We investigated the role of CardioCell in promoting angiogenesis and relieving hindlimb ischemia. Our results confirm the healing effect of CardioCell and strongly support the use of the WJ-MSCs in regenerative medicine.

## 1. Introduction

The results of decades-long research have shown that the therapeutic use of stem cells brings about significant improvements, even in cases of diseases incurable by other methods. Stem cell research is developing very dynamically. Their regenerative potential is used in medicine. Currently, the wide use of stem cells in therapies of many diseases is awaited with high hopes.

Wharton’s Jelly is a specialized type of connective tissue located inside the umbilical cord, from which mesenchymal stem cells (WJ-MSCs) can be isolated. WJ-MSCs express all surface markers typical for mesenchymal stem cells [1]. 

These cells, properly cultured and stimulated, are capable of differentiating towards other types of cells [2]. The publications appearing in recent years show the huge plasticity of WJ-MSCs, which have the ability to differentiate not only towards mesodermal cells, such as cardiomyocytes, but also into cells of ectodermal (neurons) and endodermal (hepatocytes) origin [3,4]. In addition, unlike ESC (Emrbyonic Stem Cells), they do not create ethical problems in scientific research. They have higher proliferation and self-renewal rate compared to somatic stem cells [4]. Wharton’s Jelly is an extremely valuable source of stem cells. It has been calculated that up to 4 × 10^5^ stem cells can be isolated from a 15 cm long umbilical cord fragment. WJ-MSCs are also characterized by high subdivision activity. They are well tolerated by the immune system, which favors the possibilities for their use in medicine, both in auto and allogeneic transplants [5].

Chronic ischemia of the lower limbs is a condition in which blood flow through the arteries supplying blood to the legs is restricted. The most common cause of narrowing of the arteries which leads to chronic ischemia of the lower limbs is atherosclerosis. Risk factors such as age; smoking; hypertension; diabetes; a high total cholesterol level and a low HDL cholesterol level; and obesity or atherosclerosis, (e.g., in the carotid arteries), significantly increase the risk of ischemic disease of the lower limbs [6].

The classical symptom of ischemic disease of the lower limbs is intermittent claudication—calf pain during walking, which subdues at rest. Some patients in more advanced states also complain of hip and buttock pain. Extreme stenosis of the arteries causes blood to not reach the tissues in sufficient quantity, which leads to ischemia. The pain becomes permanent, appears initially at night, decreases after lowering the legs, but after some time also occurs at rest, during the day and intensifies clearly during exercise [7]. Early diagnosis and treatment with non-invasive objective vascular tests, such as the ankle-brachial index (ABI) are of key importance [8].

Lower limb ischemia affects approximately 20% of adults over 50 years of age, exposing them to the negative effect of peripheral ischemia. However, half of the patients do not show signs of disease. Therefore, the actual effect of peripheral ischemia is difficult to assess, leading to ineffective treatment and deterioration of prevention effects. Interestingly, in patients with multifactorial atherosclerosis, the presence of peripheral ischemia is associated with a further increase in mortality, regardless of the classic cardiovascular risk factors. According to these observations, it has been shown that limb ischemia can remotely affect atherosclerotic vascular remodeling in coronary and cerebral circuits [9]. The presence of limb ischemia is often associated with endothelial dysfunction [10], and consequently with more severe atherosclerosis of distant regions [11]. Limb ischemia can negatively affect the remodeling of vessels in other regions, such as coronary and cerebral arteries. Severe cases of lower limb ischemia may lead to limb amputation or even death [12].

Most cases of lower limb ischemia are treated conservatively, based on the exclusion of atherosclerosis risk factors and the use of vasodilators, antiplatelet agents and blood clotting agents. Patients with significant ischemia are eligible for surgical treatment. Qualification for surgical treatment is based on the location of narrowing, reduction in the vessel’s diameter, the degree of stenosis, severity of the symptoms and inadequate effects from conservative treatment. Currently, angiologists and vascular surgeons use various methods of treatment that can be divided into endovascular procedures and “open” surgical procedures [13].

Human’s vessels can remodeled in response to various factors. It is believed that the underlying processes are angiogenesis (capillary proliferation) and arteriogenesis (enlargement of previously existing arterioles). Understanding these processes can potentially result in better treatment options for patients suffering from clinical conditions characterized by insufficient blood supply (e.g., peripheral artery disease). For this purpose, many researchers use animal model of hindlimb ischemia, in which ischemia is caused by a surgical operation [14].

Lower limb ischemia in patients is a consequence of many factors, depending on the degree and level of vasoconstriction, coexistence of other diseases and genetic background. A large number of different factors hamper development of a good animal model for the study of this disease. Animal models of hindlimb ischemia were developed in mice [15], rats [16] and rabbits [17]. The development of lateral arteries caused by ischemia and new medical therapies are usually tested on mouse models. Surgical procedures include a single ligation of the femoral artery or iliac artery [15], complete artery excision [11], and sometimes even, complete removal of the vein and nerves [12].

In the present study, we have examined the therapeutic potential of the human, WJ-MSCs derived product CardioCell for hindlimb ischemia treatment. Cardio is a medicinal preparation containing suspension of MSCs isolated from Wharton’s jelly of a umbilical cord. The product has been produced in our laboratory under the CIRCULATE project (Strategmed2/265761/10/ NCBR/2015)–“Regeneration of ischemic damages in cardiovascular system using Wharton’s jelly as an unlimited source of mesenchymal stem cells for regenerative medicine.”

We have investigated the properties of CardioCell that play crucial role in the promotion of angiogenesis and muscle regeneration. We have estimated the expression levels of proangiogenic factors and adhesion molecules responsible for homing and regeneration effect. In addition, our results supply data on the lifespan of CardioCell in damaged tissue after injection. 

## 2. Results

### 2.1. Characterization of Genetically Modified Cardio

The Cardio-0007 and Cardio-0007 Luc^+^ cells cultured under standard conditions adhere to the culture dish surface and exhibit fibroblast-like morphology (Figure 1A,B). The Cardio-0007 Luc+ show expression of luciferase. The efficiency of cell transduction with the luciferase gene was verified by measuring the luminescence signal by the Ami-Spectral Optical Imaging system (Spectral Instruments Imaging) which confirmed the effective genetic modification of the cells (Figure 1C). The functionality of the modified Cardio-0007 was also confirmed by immunocytochemical staining (Figure 1D–G).

### 2.2. The Proangiogenic Character of Cardio

In order to investigate the proangiogenic properties of the Cardio, a panel of characteristic markers has been studied: *VEGF-A* (*vascular endothelial growth factor A*), *ANGPT-1* (*angiopoietin 1*), *HGF* (*hepatocyte growth factor*), *FGF-1* (*fibroblast growth factor-1*), *FGF-2* (*fibroblast growth factor-2*), *PDGF-A* (*platelet derived growth factor subunit A*) and *MMP-1* (*matrix metalloproteinase-1*). The analysis of the genes expression by real-time PCR showed relatively high expression of these factors at the mRNA level (Figure 1I). The investigated genes were selected on the basis of transcriptomic profile (Figure 1H). The Cardio-0007 and Cardio-0007 Luc^+^ express numerous adhesion molecules (Figure 1J). The comparative analysis of surface markers expression and the transcriptome in CardioCell and CardioCell Luc^+^ did not show significant differences, which confirmed the lack of influence from genetic modification on cells’ properties (Figure 1H,J).

### 2.3. In Vivo Verification of CardioCell Modification

The Cardio-0007 Luc+ was subcutaneously injected into mice at two different doses: 3 × 10^5^ and 1 × 10^6^ cells (Figure 2A). After administration, the luminescence signal was maintained up to 7 days (Figure 2A–F). The most significant luminescence signal reduction was recorded in the first two days (Figure 2B), and then the signal gradually decreased until the 7th day, when it was undetectable (Figure 2B,F). For further experiments in the mouse model of hindlimb ischemia, a dose of 1 × 10^6^ cells was selected, based on the intensity of the signal after intramuscular administration of different doses of Cardio-0007 Luc+.

### 2.4. In Vivo Regeneration Potential of Cardio–Murine Model of Hindlimb Ischemia

To assess the influence of CardioCell on muscles regeneration in hind limbs, a study lasting 21 days was designed. The scheme of the study shows the plan according to which the in vivo experiment was performed (Figure 3A,B). Mice were divided into three groups: Cardio-0007 Luc^+^ (injected with the cells), PBS (injected with PBS) and SHAM (sham operated). In all groups, both functional and behavioral observations were performed (Figure 3C,D; Figure 4A–H). The Cardio-0007 Luc^+^ was administrated into mice subcutaneously at a concentration of 1 × 10^6^ cells. The luminescence signal was detected up to 14 days after injection (Figure 4A–H). Measurement of blood flow before and shortly after the surgery showed high efficiency of the surgical procedure in ischemia induction. The blood flow in operated limb decreased to about 70% compared to the healthy limb (Figure 3C,D).

During the 21 day-long experiment we observed that the blood flow in Cardio-treated mice’s group raised compared with controls (sham and PBS groups) (Figure 4H). We observed no health deterioration in the mice. We also did not notice any visible changes in the site of the cells’ implantation. The point score evaluating the physical, functional and motor condition of the mice’s hind limbs also confirmed an improvement in the group with transplanted cells compared to the control groups (Figure 4G,H).

The histological analysis of ischemic muscles also showed positive changes in their structure. In the muscle of the CardioCell group, numerous cells infiltrated; proliferating cells are visible, which indicates tissue regeneration. In the PBS-group, a large amount of calcification appeared. In the CardioCell group, the calcifications were also noticeable, but in much smaller quantities compared to PBS group (Figure 5).

## 3. Discussion

Peripheral artery disease is often one of the complications in patients suffering from diabetes. Furthermore, diabetic patients are prone to develop critical limb ischemia (CLI) that can lead to limb amputation [18,19]. Over the last few decades, stem cells-based therapy has risen as an attractive alternative method to traditional surgical treatment.

Mesenchymal stem cells are the main cell type used in regenerative medicine. Due to their biological activity, mainly due to the secretion of proangiogenic factors and their ability to differentiate into other cell types, MSCs can contribute to tissue recovery from injury [20]. These properties make MSCs the preferable stem cells type to treat some diseases, especially ischemic diseases. In the present study, we used Cardio, a product derived from Wharton’s Jelly of umbilical cords. As we have shown in our previous paper, the WJ-MSCs are ideal candidates for cell therapy, as they express high levels of pluripotency markers (OCT-3/4, Nanog) which confirms their stemness, and, what is very important, is that they do not possess protumorigenic properties [21].

Our transcriptomic analysis has shown that CardioCell express great number of genes involved in the angiogenesis process. The high expression of angiogenic growth factors, such as VEGF, HGF or FGF stimulates blood vessel development [22,23]. VEGF as a key mediator of angiogenesis; it facilitates the formation of new blood vessels, the remodeling process, and stimulates mitogenesis and survival the of endothelial cells [24,25]. It should be noted that angiogenesis is essential for wound healing; it results in tissue regeneration. The main advantage of stem cells from Wharton’s Jelly is their ability to stimulate endogenous angiogenesis, thanks to the paracrine mechanism, by which WJ-MSCs secrete numerous growth factors [25]. This was also confirmed by our transcriptomic studies as well as by Real-Time PCR on selected proangiogenic factors.

The success of stem cell therapy relies on precise homing and engraftment of the cells to target ischemic tissue. Cell interactions are mediated by a set of adhesion molecules present on cell surface. Our evaluation of surface proteins by use of the Lyoplate screening technique showed that CardioCell expresses a broad panel of molecules connected with adhesion, especially VLA molecules that are also involved in cell migration.

Cell labeling is a very important element in assessing the ability of cells to engraft in a living organism. Transplanted cells can be monitored for their migration, viability capacity and behavior in in vivo conditions [26]. Recently, many research groups showed the methods of tracking transplanted cells to mice [27], and thus the understanding of the interaction of the transplanted cells with recipient’s cells [28]. In vivo-tracking of the transplanted cells gives the opportunity to obtain reliable information and results of the applied therapy. In many animal studies, a magnetic resonance imaging method is used for this purpose [29,30,31,32]. Methods of genetic cell modification, fluorescent dye loading, BrdU or isotype labeling techniques, and many others, have also been successfully used [33]. Our research shows the possibility of genetic modification of cells and imaging in vivo transplanted cells by measuring the luminescence signal. Using modified WJ-MSCs expressing the luciferase gene allowed to visualize in vivo localization and determine the viability of the transplanted cells. Cardio-0007 and Cardio-0007 Luc+ exhibit identical features, both morphologically and molecularly. They have identical fibroblast-like morphology and surface-antigen expression. Wide transcriptome profile analysis has also not revealed significant discrepancies of gene expression in the mRNA levels. The high efficiency of cell labeling was confirmed by immunocytochemical staining and measurement of luminescence signal.

Animal models of diseases are a desirable tool for assessing the effectiveness of a therapeutic strategy, because no in vitro tests are able to reflect the complexity of processes taking place in the body. Understanding of angiogenesis and arteriogenesis processes may result in better therapeutic options for patients suffering from lower limb ischemia. For this purpose, many researchers use the animal model of hindlimb ischemia, in which ischemia is induced surgically. Surgical procedures extend from a single ligation of the femoral artery [16,17] or iliac artery [14] to complete artery dissection [15], and sometimes even veins’ and nerves’ excision [11]. Mouse hindlimb ischemia induced by surgical operation (i.e., artery ligation) is a useful model to study the angiogenic properties of MSCs [21,34]. There are factors, such as age and weight, which should be taken into consideration when selecting the most appropriate model for a given study. Several models of induced ischemia were described [34]. In our study, we adapted the model involving proximal and distal ligation of the femoral artery, which results in almost completely restricted blood flow to the thigh muscle. The protocol of murine model of hindlimb ischemia developed by us is characterized by a large decrease in blood flow in the operated limb (large ischemia measured immediately after surgery) and in some cases with necrosis of the nails [32]. However, we did not observe any necrotic changes in the feet or the whole limb, like in the case of some hindlimb ischemia induction protocols [11], sometimes ending in the necessity to amputate the limb [32,35]. Therefore, we were able to control and analyze changes in the blood flow levels in the limbs throughout the entire experiment.

Histological analysis revealed that ischemia induced in our model showed extensive muscle injury with signs of tissue damage, including of the presence of fibrosis and inflammatory cell infiltration [36,37]. A deposition of connective tissue is associated with post-injury tissue repair, resulting in the loss of the tissue function [11,12]. H&E staining revealed visible differences in muscle structure between CardioCell and PBS groups of mice. Each mouse used in experiments was examined in terms of blood flow, behavior and structure of muscles (histological analysis). The behavioral observations correlated with blood flow assessment and histological analyses of muscle tissue. We simultaneously evaluated blood flow (before and immediately after the surgery and during the experiment until 21st day) and gait.

After CardioCell administration we noticed symptoms of tissue regeneration at the site of injury. We observed decreased fibrosis and a higher number of proliferating cells, whereas in control group mice that received PBS, calcification appeared. Cell infiltration alone is not a tissue regeneration process, but it is a process of cleansing the tissue from dead cells after damage, which is not associated with xenograft rejection. Infiltrating cells are mainly neutrophils and macrophages. The lymphocyte population, in turn, does not occur because the NOD-SCID immune deficiency mice used in the experiment are characterized by an absence of functional T cells and B cells, lymphopenia, hypogammaglobulinemia and a normal hematopoietic microenvironment. NOD-SCID mice accept allogeneic and xenogeneic grafts, making them a suitable model for cell transfer experiments. The improvement in blood flow was not associated with inflammatory process in the course of xenograft rejection because the immune system of the NOD-SCID mice used in the experiment do not have a lymphocyte population that could be responsible for transplant rejection. Macrophages and neutrophils are not able to reject transplanted WJ-MSCs. Cell infiltration was also observed in the PBS group; therefore, it was not a symptom of xenograft rejection.

The analysis of measured parameters taken into account in our experiments (i.e., blood flow, gait and muscle structure) do not constitute a direct relationship between the induction of hindlimb ischemia model itself. The main aim of the study was to demonstrate the proangiogenic properties of mesenchymal stem cells isolated from Wharton’s Jelly, and in consequence their regenerative capacity in a mouse model of hindlimb ischemia. Therefore, parameters determining gait ability, the level of blood flow and histological analysis were evaluated during the experiments. All the obtained results agreed with each other. In the mouse group after the cells’ administration, the gait improvement was accompanied by a rise in blood flow. The influence of CardioCell on muscle regeneration has been estimated on the basis of a comparison analysis of all examined parameters (i.e., blood flow, gait and histology). Our results clearly showed that blood flow in damaged limb gradually increased, which undoubtedly indicated a therapeutic effect of MSCs therapy.

In our study, we wanted to determine the regenerative potential of CardioCell in a mouse model of hindlimb ischemia. For the purpose of our modified protocol of ischemia induction of the hindlimb, we created a score scale (modified ischemia score) assessing the functionality and behavior of the operated limbs during the experiment. The Tarlov scale, which has been widely used in many studies on the hindlimb ischemia model [32], did not correspond to our observations. In the CardioCell Luc^+^ group, we observed a significant improvement in the functionality of the limbs, which was reflected in the increased level of blood flow compared to the PBS group. The mice from CardioCell group recovered their ability to walk normally.

The duration of the experiment depends on many factors that determine the therapeutic window, and varied from about 7 to 28 days [11]. The most important factors were the genetic background of mice, their age and the method used to carry out the surgical procedure [38]. Another issue is that collateral vessels form in the mice quickly, which limits the therapeutic window for potential therapeutic agents [39]. Based on our own pilot studies and analyzing the data described in the literature [10,35,40], we created the experimental scheme that lasts 21 days.

In order to evaluate the hindlimb ischemia model, blood perfusion imaging is commonly used [41]. A laser Doppler imager was used to assess limb perfusion by acquiring flow images of the ischemic and the contralateral foot [10,17,32,41]. In laser-Doppler devices, by which the image of tissue perfusion is obtained, the image is created slowly due to the limitations of this technology; i.e., the perfusion image is measured point by point and the tissue perfusion image received is composed of separate measurement points obtained at different times [42]. At the same time, that causes an additional impediment of the measurement, because it is necessary to keep the examined animal motionless to avoid artifacts [43]. In the case of the laser speckle technique, the perfusion measurement is performed for the whole image at the same time [44]. Therefore, the laser speckle systems are ideal not only for static measurements but also for dynamic ones, in which the problem of respiratory movements during intravital measurements is avoided [45].

Considering the model of the applied cell therapy, attention should be paid to the source of cells, the type (allogeneic or autologous) and the route of administration, which is directly related to the results obtained [46]. Many of the studies performed show spontaneous revascularization [14,16,32,33,35].

In our study, we observed a very promising regenerative effect of CardioCell in the mouse model of hindlimb ischemia. We observed increase of blood flow level in the ischemic limbs after administration of CardioCell compared to the PBS group and signs of muscles regeneration in the CardioCell group not present in the control group.

## 4. Materials and Methods

### 4.1. WJ-MSCs Culture

The umbilical cord used in our study was collected after caesarean section, washed with phosphate-buffered saline (PBS) solution supplemented with antibiotic-antimycotic solution and cut into small (5 mm) slices. The explants of umbilical cord were put onto the plastic flask and were cultured with special growth medium for mesenchymal stem cells (DMEM Low Glucose, Biowest, Riverside, MO, USA), supplemented with platelet lysate in a standard culture conditions under 21% of O_2_ and 5% of CO_2_ at 37 °C for 7 days. After this time, the explants were removed, and the cells were passaged using Accutase cell detachment solution (BioLegend, San Diego, CA, USA). After reaching the appropriate number of cells, the cells were cryopreserved in a freezing medium (BI Biological Industries, Cromwell, CT, USA) for further experiments.

The CardioCell was manufactured in a licensed GMP (good manufacturing practice)-compliant facility, ensuring that products were consistently produced and controlled according to quality standards. The umbilical cords were obtained as part of the project STRATEGMED II (Agreement number Strategmed2/265761/10/NCBR/2015).

An inverted light microscope Olympus IX70 with phase contrast optics, equipped with Canon EOS1100D digital photo camera, was used for routine observation, analysis and documentation of the MSCs at a morphological level.

### 4.2. The Genetic Modification of Cardio, and the Expansion and Functional Verification of Modified Cells

To modify the Cardio, the lentiviral particles were prepared accordingly to ViraPower kit protocol (ThermoFisher Scientific, Waltham, MA, USA). Transfection with calcium orthophosphate was performed four hours after seeding the HEK293T cells (9.5 × 10^6^) in suitable growth medium (DMEM High Glucose supplemented with 10% of FBS). To generate the viral particles, the transfection mixture was prepared as follows: 26–39 µg of interest plasmid DNA (PGK-V5-LUC@pLenti6/Ubc expression plasmid), 65 µL of 2.5M CaCl_2_; 650 µL of 2 × BBS (BES buffered saline, pH = 7.2); 585 µL H_2_O; and pLP1, pLP2 and pLP-VSVG packaging plasmids. Then the transfection mixture was incubated for 20 min at room temperature. A previously prepared mixture with plasmids was added to growth medium with 25 µM chloroquine (Sigma-Aldrich, St. Louis, MO, USA). The whole mixture was transferred into T75 bottle with HEK293T cells. Harvested medium with lentiviral particles was centrifuged (3000 rpm, 10 min, 4 °C).

To determine the titer of lentiviral stocks, HT1080 cells were infected. Four hours prior to transduction, 5 × 10^4^ cells were seeded on 24-well plate. Various volumes of virus solutions (0.1–100 µL) were added to seeded HT1080 cells. For 7 days, cells were selected by geneticine antibiotic and the colonies of modified (LUC-expressing) cells were counted to calculate transducing units (TU) in 1 mL of harvested medium.

To obtain modified cell lines, Cardio-0007 were cultured in optimal medium conditions and were infected using generated lentiviral particles, introducing the *LUC* gene into WJ-MSCs line. Transduction was performed with MOI = 5 (multiplicity of infection) on a T75 culture dish. The Cardio-0007 line was selected with 200 µg/mL geneticin for two weeks and verified by measuring the luminescence signal by the Ami-Spectral Optical Imaging system (Spectral Instruments Imaging). To obtain a large number of modified cells for further experiments, Cardio-0007 Luc+ cells were cultured in the Quantum Cell Expansion Bioreactor (Therumo BCT, Lakewood, CO, USA), and then cryopreserved. All analyses were performed at passage 5.

### 4.3. Immunocytochemistry

The immunocytofluorescence staining method was used to verify the functionality of the luciferase genetic modification of the Cardio-0007 Luc^+^ at the protein level. The Cardio-0007 Luc^+^ cells were plated on 24-well plates on glass coverslips. When the cells reached sufficient confluence, the culture medium was drained and the cells were washed with DPBS, and fixed with 4% paraformaldehyde for 20 min at room temperature. The next step was permeabilization of cell membranes with 0.1% Triton X-100 solution in PBS (5 min at room temperature). After three washes with PBS, the sites of non-specific antibody binding were blocked using a 3% BSA solution (in PBS) for 30 min at room temperature. Then the cells were incubated overnight at 4 °C with the appropriate primary antibody diluted in 3% BSA. The following day, the cells were washed 3 times with PBS and incubated at room temperature in the dark for 1 h with the appropriate secondary antibody diluted in 3% BSA and with Hoechst 33342 (nucleic acid binding) dye. The unbound antibody was washed three times with PBS. Stained cells were observed by fluorescence microscopy and analyzed with comparison to an isotype control to confirm the specificity of staining and exclude cells with autofluorescence.

### 4.4. RNA Isolation, Reverse Transcription and Real-Time PCR Analysis (qPCR)

Total RNA was isolated using the RNA GeneMATRIX Universal RNA Purification Kit (EURx, Gdańsk, Poland) according to the manufacturer’s instructions. Reverse transcription was performed using commercial reagent kit: M-MLV Reverse Transcriptase (Promega, Madison, WI, USA). The expression of the proangiogenic factors *VEGF-A, ANGPT-1, HGF, FGF-2* and *MMP-1* at the mRNA level were verified by qPCR using TaqMan Assay (ThermoFisher Scientific, Waltham, MA, USA) and Blank qPCR Master Mix reagent (EURx, Poland). The mRNA expression level for all samples were normalized to the housekeeping gene *GAPDH*’s transcript as a reference gene. Levels of mRNA expression were determined with the 2^−ΔΔCt^ method. The Real-time qPCR was performed with the QuantStudio 7 Flex Real-Time PCR System (Applied Biosystems, Foster City, CA, USA).

### 4.5. Lyoplate Screening Panel

The expression level of adhesion molecules was evaluated using Lyoplate technology from BD (Becton Dickinson, USA). The Cardio-0007 and Cardio-0007 Luc+ were labeled with monoclonal primary antibodies directed against selected surface antigens and subsequently incubated with secondary antibodies conjugated with fluorophore for 30 min at 4 °C in darkness. The procedure was carried out according to manufacturer’s instructions. The cells were acquired using Attune Nxt Flow Cytometer (ThermoFisher Scientific) equipped with autosampler, that allows fully automated sample acquisition from a 96-well plate. A total of 10,000 viable cells (events) were analyzed per sample. The data was analyzed by use of the Attune NxT Software v2.2 and presented in the table.

### 4.6. RNA Preparation for Whole-Genome Sequencing

Total RNA was extracted from Cardio-0007 and Cardio-0007 Luc+ samples using the QIAGEN RNeasy Mini-Kit (QIAGEN, Hilden, Germany). RNA from all samples was run on an Agilent TapeStation System (Agilent Technologies, USA) to assess quality, and to estimate RNA concentrations, RNA was quantitated by Promega QuantiFluor Dye System on Quantus Fluorometer (Promega, USA).

### 4.7. Gene Expression Quantification

300 ng of total RNA was used to generate biotinylated cRNA using the TargetAmp-Nano Labeling Kit for Illumina Expression BeadChip (Epicentre-an Illumina Company, Madison, WI, USA), which was fragmented and hybridized to an Illumina Whole Genome Expression Chip, HumanHT-12 v3.0. BeadChips were then washed and stained, and subsequently scanned to obtain fluorescence intensities. Expression profiles were generated by hybridizing 750 ng of cRNA to Illumina HumanHT-12 v3.0 BeadChips according to Illumina whole-genome gene expression direct hybridization assay guide (Illumina, San Diego, CA, USA).

### 4.8. Induction of the Mouse Model of Hindlimb Ischemia in the NOD-SCID Strain

The murine hindlimb ischemia model was performed in the 16–20-week-old males of NOD-SCID mice. The mice were anesthetized with isoflurane. Then the left hindlimb was shaved on the thigh, and the skin was cut in the middle of the thigh. The femoral artery was exposed and ligated in 4 sites using a silk thread (7-0)—2 ligations in the proximal and 2 ligations in the distal parts of the femoral artery. In addition, the arterial vessel was cut between the ligatures of the artery. The body shells were closed using polypropylene soluble threads (5-0). The effectiveness of induced ischemia was assessed by comparing the level of blood flow in the hindlimb before and shortly after the operation, using a system equipped with a laser speckle (Blood flow Imager OMEGAZONE, OMEGAWAVE, INC., Tokyo, Japan). In control groups (sham operated) mice went through all stages of the operation (narcosis, and the opening and closing of skin layers), but without artery ligation.

The experiments on animals were carried out in accordance with the guidelines for the care and use of laboratory animals. The protocol was approved by the 2nd Local Institutional Animal Care and Use Committee (IACUC) in Krakow by decision number 278/2015 on 15 December 2015 and 162/2015 on 24 June 2015.

### 4.9. Transplantation of Genetically Modified Human Cardio-0007 (Cardio-0007 Luc^+^) in a Mouse Model of Hindlimb Ischemia

Mice were divided into three research groups: the first group was comprised of operated mice transplanted with Cardio-0007 Luc+; the second group—operated mice treated with PBS; the third group consisted of control mice (sham operated). Each group involved 6 mice. The next day after surgery, the modified stem cells were transplanted into mice for therapeutic purposes by intramuscular administration. Before administration, mice were sedated with isoflurane. A suspension of the Cardio-0007 Luc^+^ (at a density of 1 × 10^6^/30 µL in saline or saline alone in the PBS group) was injected intramuscularly into three locations within the ischemic muscle of the hindlimb (after shaving and slitting the skin on the operated thigh). The administration procedure was repeated three times, at a weekly interval. In the control group, mice with ischemic hindlimbs received saline injections in the same scheme.

### 4.10. The In Vivo Visualization of Cardio-0007 Luc^+^

In order to visualize in vivo the migration, localization and determine the viability of the transplanted Cardio-0007 Luc^+^, measurements of the luminescence signal were performed by use of the Optical Preclinical Imaging System for small animals (Ami, Spectral Instruments Imaging). Before imaging of the modified cells (Cardio-0007 Luc^+^), mice received the substrate (200 μL luciferin) intraperitoneally. The luciferin was also injected in the PBS mice’s group.

### 4.11. In Vivo Evaluation of the Symptoms of Hindlimb Ischemia—Blood Flow Measurements

The symptoms of ischemia in NOD-SCID mice were evaluated on the basis of blood flow measurement using a laser speckle system. Every time the results of the blood flow measurement in the hindlimb were compared to the unoperated, healthy limb. The evaluation of the blood flow rate was performed several times during the experiment.

### 4.12. Histopathological Analysis of Muscles

Mice from each group were sacrificed after 21 days of the experiment. All the samples for histopathological analysis were harvested within the boundaries of the ischemic area. The muscle tissue was fixed in 10% formalin and embedded in paraplast. Next, the tissue was cut into semi-thin sections, and stained with hematoxylin and eosin (H&E) according to standard procedure. The muscles of the hind limbs were collected from each mouse group, both operated and healthy.

### 4.13. Statistical Analysis

Statistical analysis was performed using GraphPad Prism 7 software. Data are shown as ± standard error of the mean. Statistical comparisons of the experimental and control groups were evaluated using an analysis of two-way ANOVA tests with post-hoc Fisher’s tests, with *p* ≤ 0.0001 considered statistically significant differences.

## 5. Conclusions

Local CardioCell transplantation induces a neovascular response, resulting in a significant increase in blood flow in the ischemic limbs. CardioCell is also capable of spontaneously regenerating the muscular tissues. We demonstrated that the laser speckle imaging of ischemic muscles is very useful for evaluating the regeneration process.

The results of our study have confirmed the high therapeutic potential of CardioCell and their proangiogenic properties that promote angiogenesis. Our findings strongly support use of CardioCell in cardiovascular diseases treatment.

The results obtained from this study are the basis of a clinical trial carried out by our team with the use of the “CardioCell” product, based on WJ-MSCs in three indications: acute myocardial infarction (AMI-Study, EudraCT number: 2016-004662-25), heart failure (CIHF, EudraCT number: 2016-004683-19) and critical limb ischemia (NO CLI-Study, EudraCT number: 2016-004684-40).

## Figures and Tables

**Figure 1 ijms-20-04632-f001:**
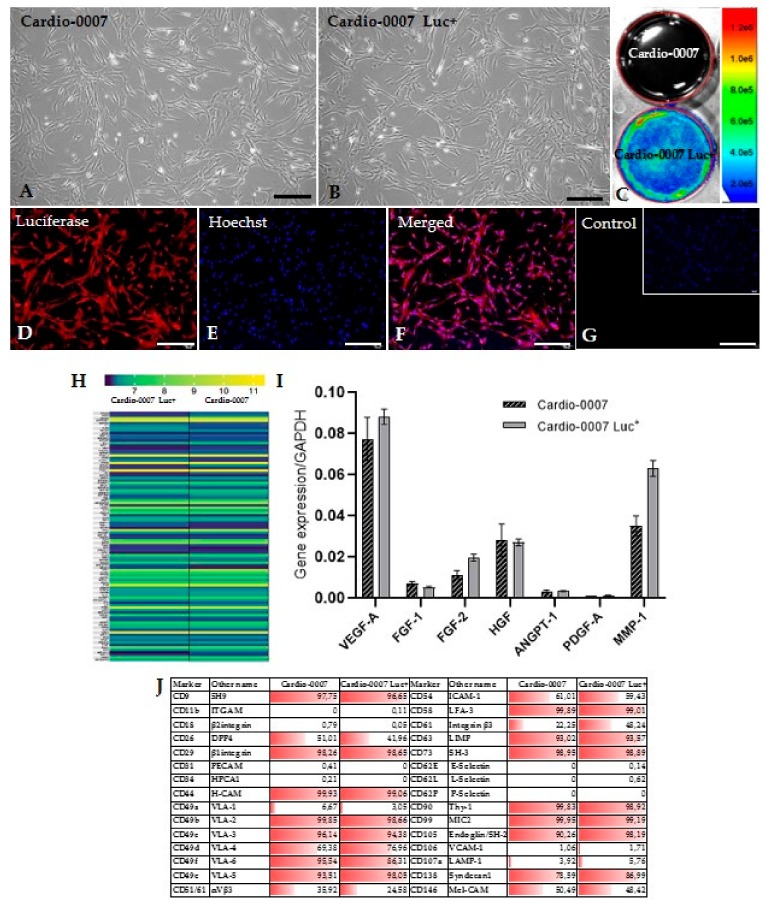
(**A**,**B**) The culture of Cardio-0007 and Cardio-0007 Luc+ (scale bar = 20µm). (**C**) The cell imaging with an AMI Imaging System. Cardio-0007 Luc+ emits a luminescence signal. (**D**–**G**) Immunofluorescence staining detecting luciferase expression in Cardio-0007 Luc+ (scale bar = 40µm). (**H**) heatmap of selected proangiogenic genes expressed in Cardio-0007 and Cardio-0007 Luc+. (**I**) expression level of selected proangiogenic factors in Cardio-0007 and Cardio-0007 Luc+. (**J**) adhesion molecules expressed (%) on Cardio-0007 and Cardio-0007 Luc+.

**Figure 2 ijms-20-04632-f002:**
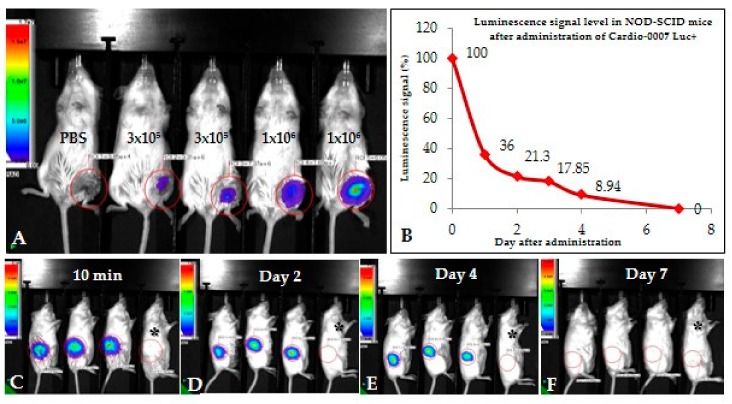
(**A**) The intensity of the luminescence signal after the administration of different doses of Cardio-0007 Luc^+^ (3 × 10^5^ and 1 × 10^6^ cells). A dose of 1 × 10^6^ cells has been selected for further experiments. (**B**–**F**) Intensity of the luminescence signal after administration of 1 × 10^6^ cells. The signal is undetectable after 7 days; black asterisk–control mouse without injected cells.

**Figure 3 ijms-20-04632-f003:**
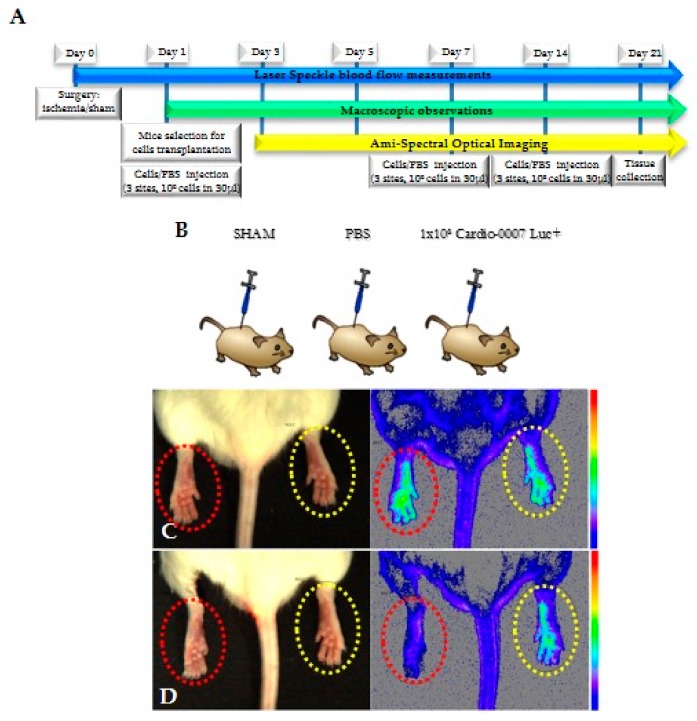
(**A**,**B**) The scheme of the experiment. (**C**,**D**) blood flow in hind limbs imaged with AMI; red circle—operated limb; yellow circle—healthy limb. Limited blood flow visible in left limb is a symptom of ischemia.

**Figure 4 ijms-20-04632-f004:**
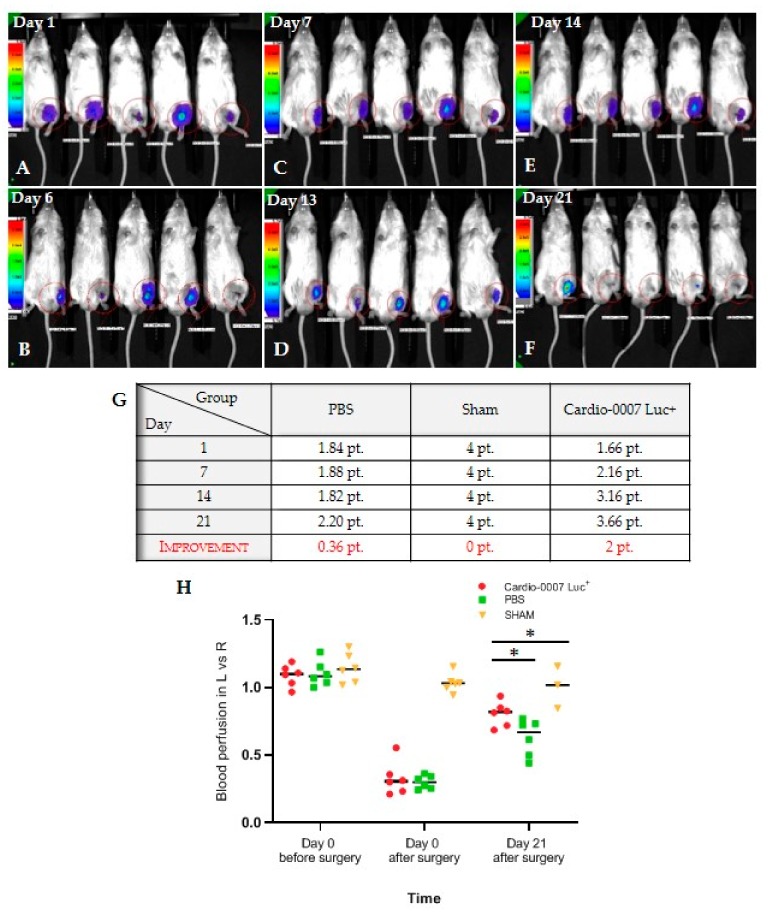
(**A**–**F)** The evaluation of the lifespan of Cardio-0007 Luc+ after administration in mice; (**G**) the results of macroscopic mice observations expressed at scale points: 0–1 pt.—pulled the hindlimb; 1.1–2 pt.—got the hindlimb stuck; 2.1–3 pt.—leaned on the hindlimb; 3.1–4 pt.—could go on hindlimb; (**H**) the results of blood perfusion measurements in the left, injured limbs (L) versus the right, healthy limbs (R). The data are in the form: mean ± SEM. * *p* ≤ 0.0001.

**Figure 5 ijms-20-04632-f005:**
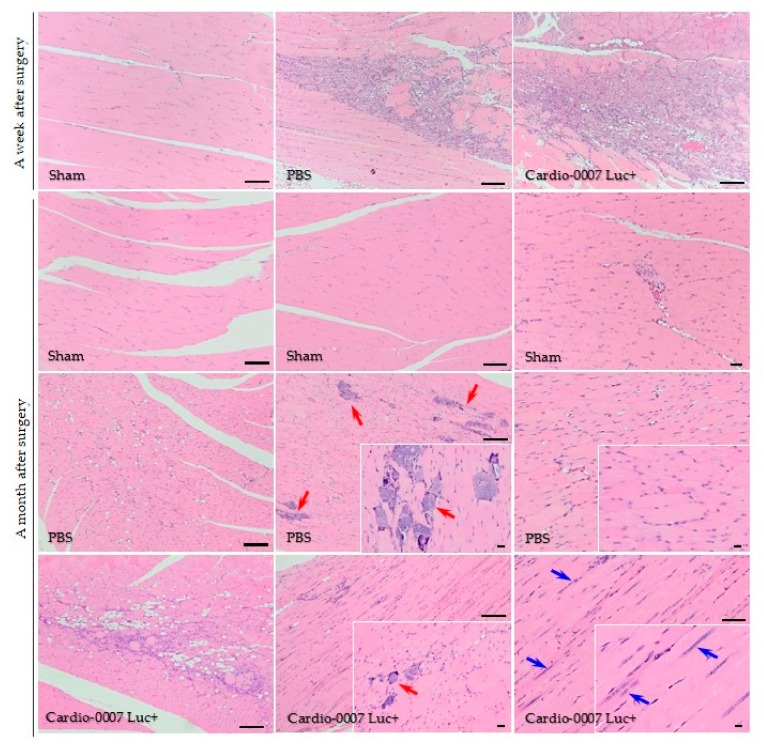
Histological assessment of the muscle’s structure after Cardio-0007 Luc+ injection; hematoxylin/eosin staining. A month after operation in the PBS mice’s group, numerous calcifications appeared (red arrows). In the Cardio-0007 Luc+ mice’s group the calcifications are rare (red arrow) and sites of tissue regeneration are visible (blue arrows); scale bar = 100 µm.

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
