# Peer review of "Regenerative Potential of the Product “CardioCell” Derived from the Wharton’s Jelly Mesenchymal Stem Cells for Treating Hindlimb Ischemia"

_ijms, 2019, doi:10.3390/ijms20184632_

Round 1
Reviewer 1 Report
This is very interesting study characterizing Wharton’s Jelly-derived mesenchymal cells (WJ-MSCs) and evaluating their potential for treatment of hind limb ischemia. This plentiful source of cells of high therapeutic potential is very attractive for regenerative medicine applications so the presented work is timely and important. Overall the paper is well written and conclusions are mostly warranted but some additional clarifications are needed.
While in vitro components characterization of cells is nicely addressing the potential regenerative properties of WJ-MSCs in vivo data are less compelling. Results are very interesting but need clarification.
Infiltration in xenograft group is interpreted as representing regeneration. In fact it is likely due to xenograft rejection. This should be discussed in more detail. And if possible addressed with new immunostainings. What is the phenotype of infiltrating cells? Staining for lymphocyte, macrophage and neutrophil markers vs. angiogenesis would help addressing this. Loss of cells at about two-three weeks after transplantation based on BLI is consistent with immunological rejection. Perivascular cuffing in Fig. 5 also is consistent with leukocyte infiltration.
Reported improvement in blood circulation could potentially be due to inflammatory processes in the course of xenograft rejection. Regardless if this is positive or negative outcome such phenomenon would not take place in case of clinical application of WJ-MSCs. This should be listed and discussed as a study limitation.
Page 5 line 139. Luminescence signal was detected for only 7 days? The figure 4 shows signal up to day 14
Author Response
Dear Reviewer,
We would like to thank you for all your comments and suggestions.
Infiltration in xenograft group is interpreted as representing regeneration. In fact it is likely due to xenograft rejection. This should be discussed in more detail. And if possible addressed with new immunostainings. What is the phenotype of infiltrating cells? Staining for lymphocyte, macrophage and neutrophil markers vs. angiogenesis would help addressing this. Loss of cells at about two-three weeks after transplantation based on BLI is consistent with immunological rejection. Perivascular cuffing in Fig. 5 also is consistent with leukocyte infiltration.
Reported improvement in blood circulation could potentially be due to inflammatory processes in the course of xenograft rejection. Regardless if this is positive or negative outcome such phenomenon would not take place in case of clinical application of WJ-MSCs. This should be listed and discussed as a study limitation.
R: We would like to thank Reviewer for this comment.
We discussed this issue in “Discussion” chapter:
Cell infiltration alone is not a tissue regeneration process, but it is a process of cleansing the tissue from dead cells after damage, which is not associated with xenograft rejection. Infiltrating cells are mainly neutrophils and macrophages. The lymphocyte population, in turn, does not occur because used in the experiment NOD-SCID immune deficiency mice are characterized by an absence of functional T cells and B cells, lymphopenia, hypogammaglobulinemia, and a normal hematopoietic microenvironment. NOD-SCID mice accept allogeneic and xenogeneic grafts making them a suitable model for cell transfer experiments. The improvement in blood flow is not associated with xenograft rejection because the immune system of the NOD-SCID mice used in the experiment does not have a lymphocyte population that could be responsible for transplant rejection. Macrophages and neutrophils are not able to reject transplanted WJ-MSCs. Cell infiltration was also observed in the PBS group, therefore it is not a symptom of xenograft rejection.
Page 5 line 139. Luminescence signal was detected for only 7 days? The figure 4 shows signal up to day 14
R: The luminescence signal after cell administration was detectable for up to 7 days (Fig. 2), therefore on this basis an experiment pattern was created according to which the cells were transplanted 3 times every 7 days (Fig.4). However, after intramuscular administration of the cells, the luminescence signal persisted for up to 14 days.
Reviewer 2 Report
The study of Musiol-Wysocka describes the therapeutic properties of Cardio product derived from human mesenchymal stem cells (MSCs) from Wharton’s jelly (WJ-MSCs). They investigated the role of Cardio in promoting angiogenesis and relieving hindlimb ischemia. The study is of interest for the scientific community. However, many aspects of the manuscript need to be extensively improved.
Major revisions
1) The use of English must be greatly improved.
Line: 98/99: In the present study, we have examined the therapeutic potential of human WJ-MSCs derive 99 Cardio product in hind limb ischemia treatment.
Line 99/100: Cardio product in hind limb ischemia treatment. We have investigated the properties of Cardio that play crucial role in promotion of angiogenesis and muscle regeneration.
Line 122/123: (…) has not shown significant differences, that confirms the lack of the genetic modification influence on cells properties (Fig. 1H, J).
Line 134/135: To assess the influence of Cardio on muscles regeneration in hind limbs, a study was designed lasting 21 days.
2) Results Fig. 5: What is the phenotype of infiltrating and proliferating cells in the Cardio group in vivo? How did the authors distinguish between calcified and non-calcified areas using only H/E and no specific stainings for calcification?
Minor revisions:
Fig. 1A, B: (…) the culture of
Fig. 2A: (…) after administration of different doses
Author Response
Dear Reviewer,
We would like to thank you for all your comments and suggestions.
Major revisions:
1) The use of English must be greatly improved.
Line: 98/99: In the present study, we have examined the therapeutic potential of human WJ-MSCs derive 99 Cardio product in hind limb ischemia treatment.
Line 99/100: Cardio product in hind limb ischemia treatment. We have investigated the properties of Cardio that play crucial role in promotion of angiogenesis and muscle regeneration.
Line 122/123: (…) has not shown significant differences, that confirms the lack of the genetic modification influence on cells properties (Fig. 1H, J).
5.Line 134/135: To assess the influence of Cardio on muscles regeneration in hind limbs, a study was designed lasting 21 days.
R: Our manuscript has been checked in terms of English. All sentences listed by Reviewer has been corrected.
2) Results Fig. 5: What is the phenotype of infiltrating and proliferating cells in the Cardio group in vivo? How did the authors distinguish between calcified and non-calcified areas using only H/E and no specific stainings for calcification?
R: The infiltrating cells are recognized as a neutrophils and macrophages (consultation with histopathologist). We used in our experiments NOD-SCID immune deficiency mice that are devoid of functional T cells and B cells, thus limfocytes are absent in infiltrating population.
The lymphocyte population, does not occur because used in the experiment NOD-SCID immune deficiency mice are characterized by an absence of functional T cells and B cells, lymphopenia, hypogammaglobulinemia, and a normal hematopoietic microenvironment. The histological preparations were thoroughly analyzed and evaluated by a qualified and experienced histopathologist, who recognized and characterized calcifications in the tissue.
Minor revisions:
1A, B: (…) the culture of
R: This part of the publication has been changed to: Fig.1A, B The culture of Cardio-0007 and Cardio-0007 Luc+.
2A: (…) after administration of different doses
R: This part of the publication has been changed to: The intensity of luminescence signal after administration of different doses.
Best regards,
Marta Kot
Reviewer 3 Report
Aleksandra Musiał-Wysocka1, Marta Kot, Maciej Sułkowski and Marcin Majka reported a manuscript entitle, “Regenerative potential of Cardio product derived from the Wharton’s Jelly Mesenchymal Stem Cells for treating hindlimb ischemia” to International Journal of Molecular Sciences.
The authors hypothesize and investigated mesenchymal stem cells, MSCs, isolated from Wharton’s jelly (WJ-MSCs) might be utilized in both cell-based therapy and vascular graft engineering to restore vascular function, thereby providing therapeutic benefits for patients. The efficacy of WJ-MSCs lies in their multipotent differentiation ability toward vascular smooth muscle cells, endothelial cells and other cell types, as well as their capacity to secrete various trophic factors,
Even though potentially this report may draw the readers” attentions and may be contributing to the literature, current format lacks plenty issues in scientific merits.
In a hindlimb model and its feasibility, the temporal and spatial limb blood flow levels should be clarified and documented.
How would Cardio-007 (Luc+) be integrated into local tissue and how have subsequent tissue perfusion improved?
Comparison with other tissue source of MSC would be more ideal in strengthen the superiority of WJ-MSCs.
Minor points
When it appears the first time, “Cardio” should be explained more properly.
Author Response
Dear Reviewer,
We would like to thank you for all your comments and suggestions.
Please find attached our response to your questions and comments.
Best regards,
Marta Kot

Round 2
Reviewer 2 Report
All my comments and suggestions have been included in the manuscript.
Author Response
We would like to thank reviewer for insightful comment.
Reviewer 3 Report
Aleksandra Musiał-Wysocka, Marta Kot, Maciej Sułkowski and Marcin Majka reported a manuscript entitle, “Regenerative potential of Cardio product derived from the Wharton’s Jelly Mesenchymal Stem Cells for treating hindlimb ischemia” to International Journal of Molecular Sciences.
The blood flow and gait improvement are investigated, and the histology of the local muscle tissue is analyzed but there is no experiment on interaction among gait, muscle histology and blood flow
More mechanistic approach of the effect of WJ-MSCs should be added to this manuscript.
Author Response
We would like to thank reviewer for insightful comment.
Each mouse used in our experiments was examined in terms of blood flow, behavior and structure of muscles (histological analysis). The behavior observations correlated with blood flow assessment and histological analysis of muscle tissue. We would like to stress that, we simultaneously evaluated blood flow (before and immediately after the surgery and during the experiment until 21st day) and gait. All the obtained results corresponded with each other. In the mice group after cells administration, the gait improvement was accompanying by raise of blood flow. The comparison analysis of all examined parameters (i.e. blood flow, gait, histology) has confirmed positive effect in WJ-MSCs mice group.
Round 3
Reviewer 3 Report
Aleksandra Musiał-Wysocka, Marta Kot, Maciej Sułkowski and Marcin Majka reported a manuscript entitle, “Regenerative potential of Cardio product derived from the Wharton’s Jelly Mesenchymal Stem Cells for treating hindlimb ischemia” to International Journal of Molecular Sciences.
As the authors cited a line in the Discussion, “Histological analysis revealed that ischemia induced in our model stimulates extensive muscle injury with signs of tissue damage including of the presence of fibrosis and inflammatory…”, it may not be able to measure the direct effects between the “gait” and tissue blood flow, especially in a longer experimental period. This part should be re-done and discussed.
Author Response
We would like to thank Reviewer for insightful comment. According to the Reviewer’s recommendations and suggestions, we added in the Discussion more detailed explanation about the raised issue (marked in red). We would like to point out that the manuscript text has been professionally checked in terms of English.